# VISALOGY: Answering Visual Analogy Questions

**Fereshteh Sadeghi**
University of Washington
fsadeghi@cs.washington.edu

**C. Lawrence Zitnick**
Microsoft Research
larryz@microsoft.com

**Ali Farhadi**
University of Washington, The Allen Institute for AI
ali@cs.washington.edu

## Abstract

In this paper, we study the problem of answering visual analogy questions. These questions take the form of *image A is to image B as image C is to what*. Answering these questions entails discovering the mapping from image A to image B and then extending the mapping to image C and searching for the image D such that the relation from A to B holds for C to D. We pose this problem as learning an embedding that encourages pairs of analogous images with similar transformations to be close together using convolutional neural networks with a quadruple Siamese architecture. We introduce a dataset of visual analogy questions in natural images, and show first results of its kind on solving analogy questions on natural images.

## 1   Introduction

Analogy is the task of mapping information from a source to a target. Analogical thinking is a crucial component in problem solving and has been regarded as a core component of cognition [1]. Analogies have been extensively explored in cognitive sciences and explained by several theories and models: shared structure [1], shared abstraction [2], identity of relation, hidden deduction [3], etc. The common two components among most theories are the discovery of a form of relation or mapping in the source and extension of the relation to the target. Such a process is very similar to the tasks in analogy questions in standardized tests such as the Scholastic Aptitude Test (SAT): *A is to B as C is to what?*

In this paper, we introduce VISALOGY to address the problem of solving visual analogy questions. Three images $I_a$, $I_b$, and $I_c$ are provided as input and a fourth image $I_d$ must be selected such that $I_a$ *is to* $I_b$ *as* $I_c$ *is to* $I_d$. This involves discovering an extendable mapping from $I_a$ to $I_b$ and then applying it to $I_c$ to find $I_d$. Estimating such a mapping for natural images using current feature spaces would require careful alignment, complex reasoning, and potentially expensive training data. Instead, we learn an embedding space where reasoning about analogies can be performed by simple vector transformations. This is in fact aligned with the traditional logical understanding of analogy as an arrow or homomorphism from source to the target.

Our goal is to learn a representation that given a set of training analogies can generalize to unseen analogies across various categories and attributes. Figure 1 shows an example visual analogy question. Answering this question entails discovering the mapping from the brown bear to the white bear (in this case a color change), applying the same mapping to the brown dog, and then searching among a set of images (the middle row in Figure 1) to find an example that respects the discovered mapping from the brown dog best. Such a mapping should ideally prefer white dogs. The bottom row shows a ranking imposed by VISALOGY.

We propose learning an embedding that encourages pairs of analogous images with similar mappings to be close together. Specifically, we learn a Convolutional Neural Network (CNN) with Siamese quadruple architecture (Figure 2) to obtain an embedding space where analogical reasoning can be

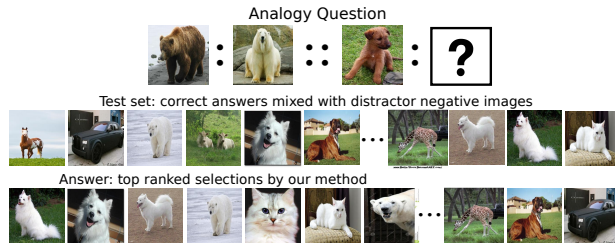

Figure 1: Visual analogy question asks for a missing image $I_d$ given three images $I_a, I_b, I_c$ in the analogy quadruple. Solving a visual analogy question entails discovering the mapping from $I_a$ to $I_b$ and applying it to $I_c$ and search among a set of images (the middle row) to find the best image for which the mapping holds. The bottom row shows an ordering of the images imposed by VISALOGY based on how likely they can be the answer to the analogy question.

done with simple vector transformations. Doing so involves fine tuning the last layers of our network so that the difference in the unit normalized activations between analogue images is similar for image pairs with similar mapping and dissimilar for those that are not. We also evaluate VISALOGY on generalization to unseen analogies. To show the benefits of the proposed method, we compare VISALOGY against competitive baselines that use standard CNNs trained for classification. Our experiments are conducted on datasets containing natural images as well as synthesized images and the results include quantitative evaluations of VISALOGY across different sizes of distractor sets. The performance in solving analogy questions is directly affected by the size of the set from which the candidate images are selected.

In this paper we study the problem of visual analogies for natural images and show the first results of its kind on solving visual analogy questions for natural images. Our proposed method learns an embedding where similarities are transferable across pairs of analogous images using a Siamese network architecture. We introduce Visual Analogy Question Answering (VAQA), a dataset of natural images that can be used to generate analogies across different objects attributes and actions of animals. We also compile a large set of analogy questions using the 3D chair dataset [4] containing analogies across viewpoint and style. Our experimental evaluations show promising results on solving visual analogy questions. We explore different kinds of analogies with various numbers of distracters, and show generalization to unseen analogies.

## 2 Related Work

The problem of solving analogy questions has been explored in NLP using word-pair connectives [5], supervised learning [6, 7, 8], distributional similarities [9], word vector representations and linguistic regularities [10], and learning by reading [11].

Solving analogy questions for diagrams and sketches has been extensively explored in AI [12]. These papers either assume simple forms of drawings [13], require an abstract representation of diagrams [14], or spatial reasoning [15]. In [16] an analogy-based framework is proposed to learn 'image filters' between a pair of images to creat an 'analogous' filtered result on a third image. Related to analogies is learning how to separate category and style properties in images, which has been studied using bilinear models [17]. In this paper, we study the problem of visual analogies for natural images possessing different semantic properties where obtaining abstract representations is extremely challenging.

Our work is also related to metric learning using deep neural networks. In [18] a convolutional network is learned in a Siamese architecture for the task of face verification. Attributes have been shown to be effective representations for semantical image understanding [19]. In [20], the relative attributes are introduced to learn a ranking function per attribute. While these methods provide an efficient feature representation to group similar objects and map similar images nearby each other in an embedding space, they do not offer a semantic space that can capture object-to-object mapping and cannot be directly used for object-to-object analogical inference. In [21] the relationships between multiple pairs of classes are modeled via analogies, which is shown to improve recognition as well as GRE textual analogy tests. In our work we learn analogies without explicity considering categories and no textual data is provided in our analogy questions.

Learning representations using both textual and visual information has also been explored using deep architectures. These representations show promising results for learning a mapping between

visual data[22] the same way that it was shown for text [23]. We differ from these methods as our objective is to directly optimized for analogy questions and our method does not use textual information.

Different forms of visual reasoning has been explored in the Question-Answering domain. Recently, the visual question answering problem has been studied in several papers [24, 25, 26, 27, 28, 29]. In [25] a method is introduced for answering several types of textual questions grounded with images while [27] proposes the task of open-ended visual question answering. In another recent approach [26], knowledge extracted from web visual data is used to answer open-domain questions. While these works all use visual reasoning to answer questions, none have considered solving *analogy* questions.

## 3  Our Approach

We pose answering a visual analogy question $I_1 : I_2 :: I_3 :?$ as the problem of discovering the mapping from image $I_1$ to image $I_2$ and searching for an image $I_4$ that has the same relation to image $I_3$ as $I_1$ to $I_2$. Specifically, we find a function $T$ (parametrized by $\theta$) that maps each pair of images $(I_1, I_2)$ to a vector $x_{12} = T(X_1, X_2; \theta)$. The goal is to solve for parameters $\theta$ such that $x_{12} \approx x_{34}$ for positive image analogies $I_1 : I_2 :: I_3 : I_4$. As we describe below, $T$ is computed using the differences in ConvNet output features between images.

### 3.1  Quadruple Siamese Network

A positive training example for our network is an analogical quadruple of images $[I_1 : I_2 :: I_3 : I_4]$ where the transformation from $I_3$ to $I_4$ is the same as that of $I_1$ to $I_2$. To be able to solve the visual analogy problem, our learned parameters $\theta$ should map these two transformations to a similar location. To formalize this, we use a contrastive loss function $L$ to measure how well $T$ is capable of placing similar transformations nearby in the embedding space and pushing dissimilar transformations apart. Given a $d$-dimensional feature vector $x$ for each pair of input images, the contrastive loss is defined as:

$$\mathcal{L}^m(x_{12}, x_{34}) = y||x_{12} - x_{34}|| + (1 - y)\max(m - ||x_{12} - x_{34}||, 0) \qquad (1)$$

where $x_{12}$ and $x_{34}$ refer to the embedding feature vector for $(I_1, I_2)$ and $(I_3, I_4)$ respectively. Label $y$ is 1 if the input quadruple $[I_1 : I_2 :: I_3 : I_4]$ is a correct analogy or 0 otherwise. Also, $m > 0$ is the margin parameter that pushes $x_{12}$ and $x_{34}$ close to each other in the embedding space if $y = 1$ and forces the distance between $x_{12}$ and $x_{34}$ in wrong analogy pairs $(y = 0)$ be bigger than $m > 0$, in the embedding space. We train our network with both correct and wrong analogy quadruples and the error is back propagated through stochastic gradient descent to adjust the network weights $\theta$. The overview of our network architecture is shown in Figure 2.

To compute the embedding vectors $x$ we use the quadruple Siamese architecture shown in Figure 2. Using this architecture, each image in the analogy quadruple is fed through a ConvNet (AlexNet [30]) with shared parameters $\theta$. The label $y$ shows whether the input quadruple is a correct analogy $(y = 1)$ or a false analogy $(y = 0)$ example. To capture the transformation between image pairs $(I_1, I_2)$ and $(I_3, I_4)$, the outputs of the last fully connected layer are subtracted. We normalize our embedding vectors to have unit L2 length, which results in the Euclidean distance being the same as the cosine distance. If $X_i$ are the outputs of the last fully connected layer in the ConvNet for image $I_i$, $x_{ij} = T(X_i, X_j; \theta)$ is computed by:

$$T(X_i, X_j; \theta) = \frac{X_i - X_j}{||X_i - X_j||} \qquad (2)$$

Using the loss function defined in Equation (1) may lead to the network overfitting. Positive analogy pairs in the training set can get pushed too close together in the embedding space during training. To overcome this problem, we consider a margin $m_P > 0$ for positive analogy quadruples. In this case, $x_{12}$ and $x_{34}$ in the positive analogy pairs will be pushed close to each other only if the distance between them is bigger than $m_P > 0$. It is clear that $0 \leq m_P \leq m_N$ should hold between the two margins.

$$\mathcal{L}^{m_P, m_N}(x_{12}, x_{34}) = y\max(||x_{12} - x_{34}|| - m_P, 0) + (1 - y)\max(m_N - ||x_{12} - x_{34}||, 0) \quad (3)$$

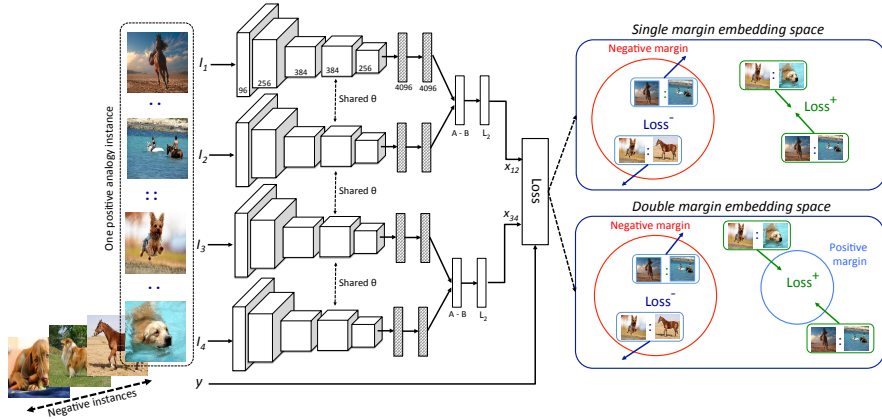

Figure 2: VISALOGY Network has quadruple Siamese architecture with shared $\theta$ parameters. The network is trained with correct analogy quadruples of images $[I_1, I_2, I_3, I_4]$ along with wrong analogy quadruples as negative samples. The contrastive loss function pushes $(I_1, I_2)$ and $(I_3, I_4)$ of correct analogies close to each other in the embedding space while forcing the distance between $(I_1, I_2)$ and $(I_3, I_4)$ in negative samples to be more than margin $m$.

## 3.2 Building Analogy Questions

For creating a dataset of visual analogy questions we assume each training image has information $(c, p)$ where $c \in C$ denotes its category and $p \in P$ denotes its property. Example properties include color, actions, and object orientation. A valid analogy quadruple should have the form:

$$[I_1^{(c_i, p_1)} : I_2^{(c_i, p_2)} :: I_3^{(c_o, p_1)} : I_4^{(c_o, p_2)}]$$

where the two input images $I_1$ and $I_2$ have the same category $c_i$, but their properties are different. That is, $I_1$ has the property $p_1$ while $I_2$ has the property $p_2$. Similarly, the output images $I_3$ and $I_4$ share the same category $c_o$ where $c_i \neq c_o$. Also, $I_3$ has the property $p_1$ while $I_4$ has the property $p_2$ and $p_1 \neq p_2$.

**Generating Positive Quadruples:** Given a set of labeled images, we construct our set of analogy types. We select two distinct categories $c, c' \in C$ and two distinct properties $p, p' \in P$ which are shared between $c$ and $c'$. Using these selections, we can build 4 different analogy types (either $c$ or $c'$ can be considered as $c_i$ and $c_o$ and similarly for $p$ and $p'$). For each analogy type (e.g. $[(c_i, p_1) : (c_i, p_2) :: (c_o, p_1) : (c_o, p_2)]$), we can generate a set of positive analogy samples by combining corresponding images. This procedure provides a large number of positive analogy pairs.

**Generating Negative Quadruples:** Using only positive samples for training the network leads to degenerate models, since the loss can be made zero by simply mapping each input image to a constant vector. Therefore, we also generate quadraples that violate the analogy rules as negative samples during training. To generate negative quadruples, we take two approaches. In the first approach, we randomly select 4 images from the whole set of training images and each time check that the generated quadruple is not a valid analogy. In the second approach, we first generate a positive analogy quadruple, then we randomly replace either of $I_3$ or $I_4$ with an improper image to break the analogy. Suppose we select $I_3$ for replacement. Then we can either randomly select an image with category $c_o$ and property $p^*$ where $p^* \neq p_1$ and $p^* \neq p_2$ or we can randomly select an image with property $p_1$ but with a category $c^*$ where $c^* \neq c_o$. The second approach generates a set of hard negatives to help improve training. During the training, we randomly sample from the whole set of possible negatives.

## 4 Experiments

**Testing Scenario and Evaluation Metric:** To evaluate the performance of our method for solving visual analogy questions, we create a set of correct analogy quadruples $[I_1 : I_2 :: I_3 :?]$ using the $(c, p)$ labels of images. Given a set $D$ of images which contain both positive and distracter images, we would like to rank each image $I_i$ in $D$ based on how well it completes the analogy. We compute the corresponding feature embeddings $x_1, x_2, x_3$, for each of the input images as well as $x_i$ for each image in $D$ and we rank based on:

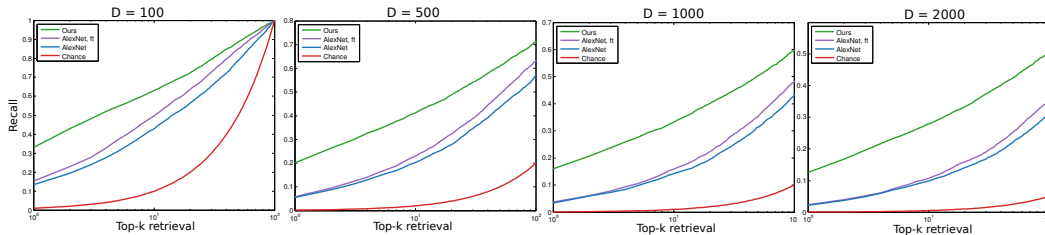

Figure 3: Quantitative evaluation (log scale) on 3D chairs dataset. Recall as a function of the number $(k)$ of images returned (Recall at top-$k$). For each question the recall at top-$k$ is either 0 or 1 and is averaged over 10,000 questions. The size of the distractor set $D$ is varied $D = [100, 500, 1000, 2000]$. 'AlexNet': AlexNet, 'AlexNet ft': AlexNet fine-tuned on chairs dataset for categorizing view-points.

$$rank_i = \frac{T(I_1, I_2).T(I_3, I_i)}{||T(I_1, I_2)||.||T(I_3, I_i)||}, \quad i \in 1, ..., n \qquad (4)$$

where $T(.)$ is the embedding obtained from our network as explained in section 3. We consider the images with the same category $c$ as of $I_3$ and the same property $p$ as of $I_2$ to be a correct retrieval and thus a positive image and the rest of the images in $\mathcal{D}$ as negative images. We compute the recall at top-$k$ to measure whether or not an image with an appropriate label has appeared in the top $k$ retrieved images.

**Baseline:** It has been shown that the output of the 7th layer in AlexNet produces high quality state-of-the-art image descriptors [30]. In each of our experiments, we compare the performance of solving visual analogy problems using the image embedding obtained from our network with the image representation of AlexNet. In practice, we pass each test image through AlexNet and our network, and extract the output from the last fully connected layer using both networks. Note that for solving general analogy questions the set of properties and categories are not known at the test time. Accordingly, our proposed network does not use any labels during training and is aimed to generalize the transformations without explictily using the label of categories and properties.

**Dataset:** To evaluate the capability of our trained network for solving analogy questions in the test scenarios explained above, we use a large dataset of 3D chairs [4] as well as a novel dataset of natural images (VAQA), that we collected for solving analogy questions on natural images.

### 4.1 Implementation Details

In all the experiments, we use stochastic gradient descent (SGD) to train our network. For initializing the weights of our network, we use the AlexNet pre-trained network for the task of large-scale object recognition (ILSVRC2012) provided by the BVLC Caffe website [31]. We fine-tune the last two fully connected layers (fc6, fc7) and the last convolutional layer (conv5) unless stated otherwise. We have also used the double margin loss function introduced in Equation 3 with $m_P = 0.2, m_N = 0.4$ which we empirically found to give the best results in a held-out validatation set. The effect of using a single margin vs. double margin loss function is also investigated in section 4.4.

### 4.2 Analogy Question Answering Using 3D Chairs

We use a large collection of 1,393 models of chairs with different styles introduced in [4]. To make the dataset, the CAD models are download from Google/Trimble 3D Warehouse and each chair style is rendered on white background from different view points. For making analogy quadruples, we use 31 different view points of each chair style which results in 1,393*31 = 43,183 synthesized images. In this dataset, we treat different styles as different *categories* and different view points as different *properties* of the images according to the explanations given in section 3.2. We randomly select 1000 styles and 16 view points for training and keep the rest for testing. We use the rest of 393 classes of chairs with 15 view points (which are completely unseen during the training) to build unseen analogy questions that test the generalization capability of our network at test time. To construct an analogy question, we randomly select two different styles and two different view points. The first part of the analogy quadruple $(I_1, I_2)$ contains two images with the same style and with two different view points. The images from the second half of the analogy quadruple $(I_3, I_4)$, have another style and $I_3$ has the same viewpoint as $I_1$ and $I_4$ has the same view point as $I_2$. Together, $I_1, I_2, I_3$ and $I_4$ build an analogy question $(I_1 : I_2 :: I_3 :?)$ where $I_4$ is the correct answer. Using this approach, the total number of positive analogies that could be used during training is $\binom{1000}{2} \times \binom{16}{2} \times 4 = 999,240$.

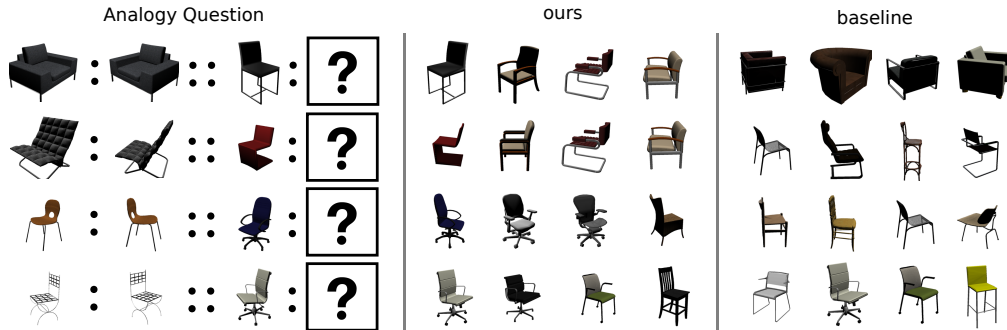

Figure 4: Left: Several examples of analogy questions from the 3D chairs dataset. In each question, the first and second chair have the same style while their view points change. The third image has the same view point as the first image but in a different style. The correct answer to each question is retrieved from a set with 100 distractors and should have the same style as the third image while its view point should be similar to the second image. Middle: Top-4 retrievals using the features obtained from our method . Right: Top-4 retrievals using AlexNet features. All retrievals are sorted from left to right

To train our network, we uniformly sampled 700,000 quadruples (of positive and negative analogies) and initialized the weights with the AlexNet pre-trained network and fine-tuned its parameters. Figure 4 shows several samples of the analogy questions (left column) used at test time and the top-4 images retrieved by our method (middle column) compared with the baseline (right column). We see that our proposed approach can retrieve images with a style similar to that of the third image and with a view-point similar to the second image while the baseline approach is biased towards retrieving chairs with a style similar to that of the first and the second image. To quantitatively compare the performance of our method with the baseline, we randomly generated 10,000 analogy questions using the test images and report the average recall at top-k retrieval while varying the number of irrelevant images ($D$) in the distractor set. Note that, since there is only one image corresponding to each (style , view-point), there is only one positive answer image for each question. The performance of chance at the top-$k$th retrieval is $\frac{k}{n}$ where $n$ is the size of $D$. The images of this dataset are synthesized and do not follow natural image statistics. Therefore, to be fair at comparing the results obtained from our network with that of the baseline (AlexNet), we fine-tune all layers of the AlexNet via a soft-max loss for categorization of different view-points and using the set of images seen during training. We then use the features obtained from the last fully connected layer (fc7) of this network to solve analogy questions. As shown in Figure 3, fine-tuning all layers of AlexNet (the violet curve referred to as 'AlexNet,ft' in the diagram) helps improve the performance of the baseline. However, the recall of our network still outperforms it with a large margin.

## 4.3   Analogy Question Answering using VAQA Dataset

As explained in section 3.2, to construct a natural image analogy dataset we need to have images of numerous object categories with distinguishable properties. We also need to have these properties be shared amongst object categories so that we can make valid analogy quadruples using the $(c, p)$ labels. In natural images, we consider the *property* of an object to be either the action that it is doing (for animate objects) or its attribute (for both animate and non-animate objects). Unfortunately, we found that current datasets have a sparse number of object properties per class, which restricts the number of possible analogy questions. For instance, many action datasets are human centric, and do not have analogous actions for animals. As a result, we collected our own dataset VAQA for solving visual analogy questions.

**Data collection:** We considered a list of 'attributes' and 'actions' along with a list of common objects and paired them to make a list of $(c, p)$ labels for collecting images. Out of this list, we removed $(c, p)$ combinations that are not common in the real world (e.g. (horse,blue) is not common in the real world though there might be synthesized images of 'blue horse' in the web). We used the remaining list of labels to query Google Image Search with phrases made from concatenation of word $c$ and $p$ and downloaded 100 images for each phrase. The images are manually verified to contain the concept of interest. However, we did not pose any restriction about the view-point of the objects. After the pruning step, there exists around 70 images per category with a total of 7,500 images. The VAQA dataset consists of images corresponding to 112 phrases which are made out of 14 different categories and 22 properties. Using the shared properties amongst categories we can build 756 types of analogies. In our experiments, we used over 700,000 analogy questions for training our network.

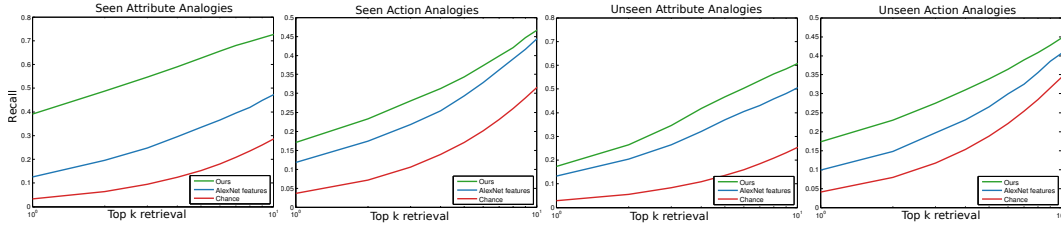

Figure 5: Quantitative evaluation (log scale) on the VAQA dataset using 'attribute' and 'action' analogy questions. Recall as a function of the number ($k$) of images returned (Recall at top-$k$). For each question the recall at top-k is averaged over 10,000 questions. The size of the distractor set is fixed at 250 in all experiments. Results shown for analogy types seen in training are shown in the left two plots, and for analogy types not seen in training in the two right plots.

**Attribute analogy:** Following the procedure explained in Section 3.2 we build positive and negative quadruples to train our network. To be able to test the generalization of the learned embeddings for solving analogy question types that are not seen during training, we randomly select 18 attribute analogy types and remove samples of them from the training set of analogies. Using the remaining analogy types, we sampled a total of 700,000 quadruples (positive and negative) that are used to train the network.

**Action analogy:** Similarly, we trained our network to learn action analogies. For the generalization test, we remove 12 randomly selected analogy types and make the training quadruples using the remaining types. We sampled 700,000 quadruples (positive and negative) to train the network.

**Evaluation on VAQA:** Using the unseen images during the training, we make analogy quadruples to test the trained networks for the 'attribute' and 'action' analogies. For evaluating the specification and generalization of our trained network we generate analogy quadruples in two scenarios of 'seen' and 'unseen' analogies using the analogy types seen during training and the ones in the withheld sets respectively. In each of these scenarios, we generated 10,000 analogy questions and report the average recall at top-$k$. For each question $[I_1 : I_2 :: I_3 :?]$, images that have property $p$ equal to that of $I_2$ and category $c$ equal to $I_3$ are considered as correct answers. The result is around 4 positive images for each question and we fix the distracter set to have 250 negative images for each question. Given the small size of our distracter set, we report the average recall at top-10. The obtained results in different scenarios as summarized in Figure 5. In all the cases, our method outperforms the baseline.

Other than training separate networks for 'attribute' and 'action' analogies, we trained and tested our network with a combined set of analogy questions and obtained promising results with a gap of 5% compared to our baseline on the top-5 retrievals of the seen analogy questions. Note that our current dataset only has one property label per image (either for 'attribute' or 'action'). Thus, a negative analogy for one property may be positive for the other. A more thorough analysis would require multi-property data, which we leave for future work.

**Qualitative Analysis:** Figure 6, shows examples of attribute analogy questions that are used for evaluating our network along with the top five retrieved images obtained from our method and the baseline method. As explained above, during the data collection we only prune out images that do not contain the $(c, p)$ of interest. Also, we do not pose any restriction for generating positive quadruples such as restricting the objects to have similar pose or having the same number of objects of interest in the quadruples. However, as can be seen in Figure 6 our network had been able to *implicitly* learn to generalize the *count* of objects. For example, in the first row of Figure 6, an image pair is ['dog swimming' : 'dog standing'] and the second part of the analogy has an image of 'multiple horses swimming'. Given this analogy question as input, our network has retrieved images with multiple 'standing horses' in the top five retrievals.

## 4.4 Ablation Study

In this section, we investigate the effect of training the network with double margins $(m_P, m_N)$ for positive and negative analogy quadruples compared with only using one single margin for negative quadruples. We perform an ablation experiment where we compare the performance of the network at top-$k$ retrieval while being trained using either of the loss functions explained in Section 4. Also, in two different scenarios, we either fine-tune only the top fully connected layers fc6 and fc7 (re-

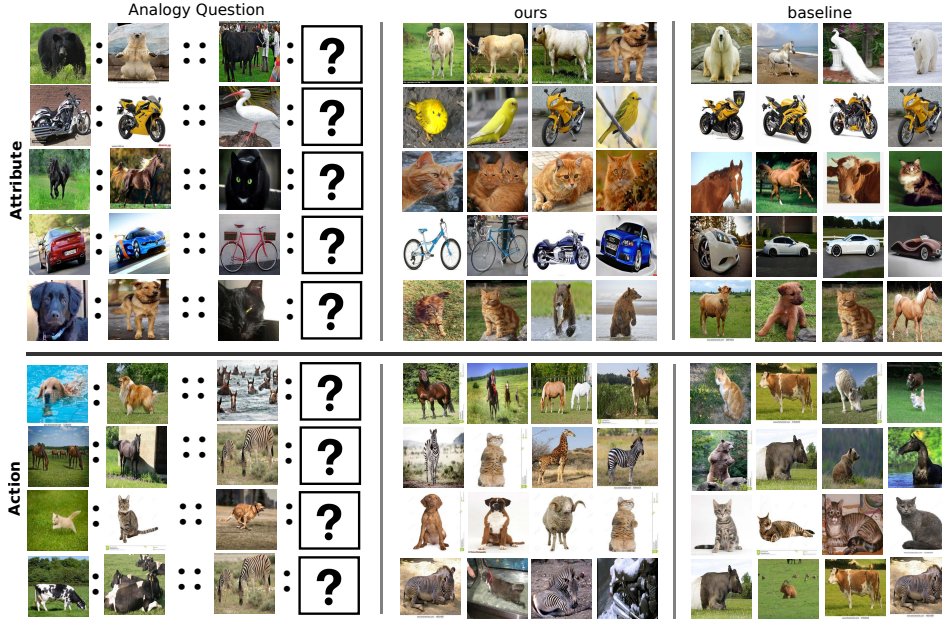

Figure 6: Left: Samples of test analogy questions from VAQA dataset. Middle: Top-4 retrievals using the features obtained from our method. Right: Top-4 retrievals using AlexNet features.

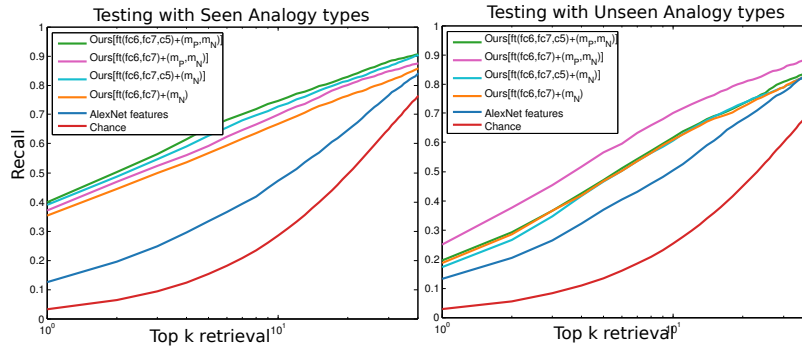

Figure 7: Quantitative comparison for the effect of using double margin vs. single margin for training the VISALOGY network.

ferred to as 'ft(fc6,fc7)' in Figure 7) or the top fully connected layers plus the last convolutional layer c5 (referred to as 'ft(fc6,fc7,c5)') in Figure 7). We use a fixed training sample set consisting of 700,000 quadruples generated from the VAQA dataset in this experiment. In each case, we test the trained network using samples coming from the set of analogy questions whose types are seen/unseen during the training. As can be seen from Figure 7, using double margins $(m_P, m_N)$ in the loss function has resulted in better performance in both testing scenarios. While using double margins results in a small increase in the 'seen analogy types' testing scenario, it has considerably increased the recall when the network was tested with 'unseen analogy types'. This demonstrates that the use of double margins helps generalization.

# 5 Conclusion

In this work, we introduce the new task of solving visual analogy questions. For exploring the task of visual analogy questions we provide a new dataset of natural images called VAQA. We answer the questions using a Siamese ConvNet architecture that provides an image embedding that maps together pairs of images that share similar property differences. We have demonstrated the performance of our proposed network using two datasets and have shown that our network can provide an effective feature representation for solving analogy problems compared to state-of-the-art image representations.

**Acknowledgments:** This work was in part supported by ONR N00014-13-1-0720, NSF IIS-1218683, NSF IIS-IIS- 1338054, and Allen Distinguished Investigator Award.

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
