[Reviews · NeurIPS 2015]

Submitted by Assigned_Reviewer_1

The paper presents a model for visual analogy retrieval. For a query image one wants to find a second image which relates to the query in the same way as the relationship between a given pair of images. More precisely, an offset pair (i1, i2) defines the analogy. For a query i3 an image i4 is to be found such that i3 relates to i4 in the same way as i1 relates to i2.

The model is formulated as an embedder, which embeds all four images ik -> xk (k = 1, ..., 4) such that (x4 - x3) = (x2 - x1). The embedder is a CNN based on AlexNet. The loss is a hinge loss.

In addition, the authors present a dataset of visual analogy.

The paper is well written and presents a well defined model which is empirically evaluate in two datasets.

My main concern with this work is that the presented problem might be a bit contrived. The authors define an analogy as relationship between images defined by different properties of these images. The philosophy of analogies being a pair of two properties is incorporated in both of their datasets.

So if an analogy is defined as pair of properties, I natural baseline would be to set up a classification problem for these properties. Then if analogy is defined by the properties of two images, then one can apply the trained classifier to estimate these properties. A second classifier would determine the class of the query image. Then the analogous image would be one of the class of the query and the missing of the two properties. Such a baseline is based on two classifiers and would be simple to set up.
Summary: This is well-written paper with an interesting problem. Although the proposed model seems to work well, there are simpler and more off-the shelf ways of addressing the problem, which were not explored. Hence, I am leaning towards waiting for a stronger justification.

Submitted by Assigned_Reviewer_2

Summary: This paper introduces a method and dataset for answering visual analogy questions (image A is similar to image B, as C is similar to D). The method is based on embedding images into a space where the distance A-B will be similar to C-D.

Quality: In general this is a high quality paper that tackles a creative and interesting problem, uses a simple but reasonable technical algorithm, and contains experiments that adequately support the claims of the paper.

Clarity: The paper is clearly written and well organized

Originality: The main idea of the paper is creative and novel (to my knowledge)

Significance: The one possible weakness is that it isn't obvious that there would be some usefulness in being able to solve visual analogy questions.

The notion makes less intuitive sense than the counterpart of word-based analogies (SAT questions).

These are more so a question of demonstrating intelligence, where the person must infer the similarity and difference between A and B and extrapolate it to another pair.

By contrast, in the experimental results, the datasets constructed represent a more restricted setting where (on each dataset individually) one already knows A & C are the same object class and B & D are similar according to some pre-existing type of attribute (on one dataset B&D always have the same viewpoint, on another they always have similar visual attributes, and a 3rd they have similar action).

An experiment on a combined dataset would somewhat strengthen results.
Summary: This paper has all the ingredients of a good paper (interesting and novel problem, good technical details, good experimental results, and good clarity).

The one weakness is that it isn't obvious what is the practical value of answering visual analogy questions.

Author Feedback
Author rebuttal: We thank reviewers for acknowledging the novelty and interestingness of our paper as well as our results. (R2: "well-written paper with an interesting problem, the proposed model works well"; R4: "The results are convincing, this is an interesting problem"; R5: "This paper treats a new problem"; R6: "The paper achieves convincing results"; R7: "This paper has all the ingredients of a good paper, main idea of the paper is creative and novel" R8: "Extensive experiments are done"). We appreciate reviewers insightful comments; we will incorporate them in any final version of the paper.

R2,R6: Two-classifier and attribute-based baseline

Direct application of the suggested two-classifier baseline is not appropriate as we explain below. We run a modified version of it and the obtained results show that our method outperforms the suggested baseline.

**Issues with the suggested two-classifier baseline:

***Unseen category/property during training:
The suggested two-classifier baseline directly uses the category/property labels to train classifiers. For solving general analogy questions, the set of properties and categories are not known at test time (Line 261, Figs 3&5). Therefore, we propose to learn a network that can generalize the transformations without explicitly using the label of categories/properties. We have produced the positive analogy pairs using the category and property labels, but we are not directly using any of those labels during training and do not optimize any softmax loss. Our loss function only optimizes the similarity between transformations.

***Extendability to questions with multiple formats:
The suggested two-classifier baseline should be aware of the format of questions (whether property is changing from A to B or the category). Otherwise, a set of rules should be devised such that the two-classifier baseline would first detect the type of question by comparing A and B, and then retrieve the appropriate image. The setup of such rules in the two-classifier baseline would become complicated and non-applicable in images with multiple property types.

**Supporting experiment for the two-classifier baseline:
We ran a new experiment on 3D chairs dataset that has analogy questions of both formats (A and B with same style but different poses as well as A and B with with same pose and different styles).

***Generalization:
The results show that our network (trained on only one format of analogy questions) is able to answer another format of the analogy test questions. In this new experiment our method outperformed AlexNet baseline with a gap of ~12%.

***Poor performance of suggested two-classifier baseline:
As the format of analogy questions is unknown, we computed the style and pose probability vectors and first detect the type of question based on the pose and style similarity between A and B and then retrieve the image with the highest pose and style similarity to images C and B accordingly. This baseline resulted in a poor performance (~30% gap) compared to our method. We will add this experiment in the final version of the paper.

R4, R6, Missing citations
Thanks for the suggestions. We will include them.

R6: People agreement on analogy questions
This is a great suggestion. We will include this in our future work.

R7: Experiment on a combined dataset
Thanks for the good suggestion. We trained and tested our network with a combined set of analogy questions (action and attribute) and obtained promising results with a gap of 5% compared with our baseline (AlexNet). The images in our natural image dataset only have one property label (either attribute or action); and there is a chance that the negative analogy pair of one question type, be the positive of another one. We will reflect this in the final version of the paper.

R7: Practical values
We agree that it does not have a direct application but it is an important problem that reflects on image understanding. Reasoning based on analogies is the first step toward deeper understanding of generalization. This paper is one step toward this challenging goal. To show our generalization capability in answering multiple formats of analogy questions we ran a new experiment (see experiment above).

R8: Analogy question format with A, B from different categories but same property
Thanks for the good suggestion. To test the generalization capability of our network to answer multiple question formats, we tested on a new set of question on 3D chairs dataset with both question types and our network obtained 12% improvement over the AlexNet baseline (see experiment above).

R8: Lack of contribution
Our contribution is to propose and solve a new problem as well as a new dataset. We demonstrated results with extensive experiments. The goal we are pursuing is "learning to generalize" which is important in image understanding. The novelty and creativity of our proposed problem is also acknowledged by R2,4,5,6,7.